# Bipartite Graph Attention-based Clustering for Large-scale scRNA-seq Data

Zhuomin Liang [1]  Liang Bai [1]  Xian Yang [2]

## Abstract

scRNA-seq clustering is a critical task for analyzing single-cell RNA sequencing (scRNA-seq) data, as it groups cells with similar gene expression profiles. Transformers, as powerful foundational models, have been applied to scRNA-seq clustering. Their self-attention mechanism automatically assigns higher attention weights to cells within the same cluster, enhancing the distinction between clusters. Existing methods for scRNA-seq clustering, such as graph transformer-based models, treat each cell as a token in a sequence. Their computational and space complexities are $\mathcal{O}(n^2)$ with respect to the number of cells, limiting their applicability to large-scale scRNA-seq datasets. To address this challenge, we propose a Bipartite Graph Transformer-based clustering model (BGFormer) for scRNA-seq data. We introduce a set of learnable anchor tokens as shared reference points to represent the entire dataset. A bipartite graph attention mechanism is introduced to learn the similarity between cells and anchor tokens, bringing cells of the same class closer together in the embedding space. BGFormer achieves linear computational complexity with respect to the number of cells, making it scalable to large datasets. Experimental results on multiple large-scale scRNA-seq datasets demonstrate the effectiveness and scalability of BGFormer.

## 1. Introduction

scRNA-seq clustering is the a fundamental step in single-cell RNA sequencing (scRNA-seq) data analysis. By grouping cells with similar transcriptional profiles, researchers can identify distinct cell types and subpopulations (Lu et al., 2023), and explore cellular heterogeneity (Zheng et al.,

[1]Institute of Intelligent Information Processing, Shanxi University, Taiyuan, China [2]Alliance Manchester Business School, The University of Manchester, Manchester, UK. Correspondence to: Liang Bai <bailiang@sxu.edu.cn>.

*Proceedings of the 43rd International Conference on Machine Learning*, Seoul, South Korea. PMLR 306, 2026. Copyright 2026 by the author(s).

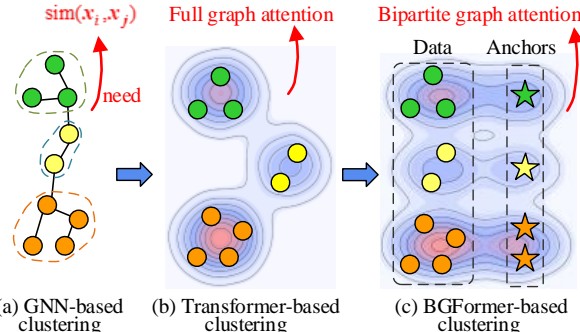

*Figure 1.* Comparison of different clustering methods.

2017). However, the high dimensionality, sparsity, and technical variability inherent in scRNA-seq data pose significant challenges to existing clustering methods.

Graph-based clustering methods provide an effective solution to these challenges by leveraging cell relationships to group similar cells into clusters (Wang et al., 2021). Both Graph Neural Networks (GNNs) and Transformers, which are powerful tools for modeling relationships, have been employed for scRNA-seq clustering. GNN-based methods rely on explicit relationships to guide the encoding of cell information (Wang et al., 2021; Zhang et al., 2025). As illustrated in Fig. 1(a), these methods require calculating the similarity among cells to construct the k-nearest neighbor (kNN) graph, as the relationships between cells are not inherently available. However, the high dimensionality and sparsity of scRNA-seq data often undermine the effectiveness of similarity metrics, limiting the performance of these clustering methods. In contrast, Transformer-based methods can better handle these challenges due to their ability to capture complex relationships without relying on traditional similarity metrics. These methods model the input as a sequence of tokens, with each cell treated as an individual token (Szałata et al., 2024; Fan et al., 2024). As shown in Fig. 1(b), the self-attention mechanism learns a full graph attention to implicitly capture the similarities among cells. By adaptively adjusting the similarity matrix, the differences between clusters are increased, making it easier to identify cell groups without explicit guidance. Although these methods eliminate the need for kNN graph construction, the quadratic computational complexity of self-attention with respect to the number of cells makes them prohibitively

expensive for large-scale datasets.

We propose a novel **B**ipartite **G**raph Trans**Former**-based clustering model (BGFormer) for large-scale scRNA-seq data. The core of BGFormer is to shift the similarity calculation for clustering from between cells to between cells and learnable anchor tokens, as illustrated in Fig. 1(c). These anchors serve as reference points for the entire dataset, allowing similar cells to aggregate similar global information. Since its number is much smaller than the number of cells, the computational complexity is reduced from $\mathcal{O}(n^2)$ to $\mathcal{O}(n)$, where $n$ is the number of cells. The anchor tokens are shared across mini-batches, making BGFormer inherently suitable for training on large-scale scRNA-seq datasets. The main contributions of this paper are as follows:

- We introduce an efficient clustering method for large-scale scRNA-seq data, achieving linear computational complexity with respect to the number of cells.

- We introduce a bipartite graph attention mechanism that learns similarity for clustering from the relationships between cells and anchor tokens.

- Experiments on large-scale scRNA-seq datasets demonstrate that BGFormer achieves lower computational cost and superior clustering performance.

## 2. Related Work

### 2.1. Graph-based scRNA-seq Clustering

Graph-based clustering methods are widely used for scRNA-seq analysis, as they model relationships among cells. Since such relationships are not directly available, these methods rely on similarity metrics to construct explicit graph structures, such as those used in k-means (McQueen, 1967) algorithms. Methods like Louvain and Leiden (Traag et al., 2019) are then applied to identify cell clusters. scICE (Kim et al., 2025) further enhance the reliability of clustering by evaluating multi-cluster label consistency. With the development of graph neural networks (GNNs) (Kipf & Welling, 2017), many methods group cells by bringing similar cells closer in the embedding space. Methods including scGAE (Luo et al., 2021), scGCL (Xiong et al., 2023), scTAG (Yu et al., 2022), and CCST (Li et al., 2022) adopt autoencoder or contrastive learning frameworks to learn cell embeddings in an unsupervised manner. scG-cluster (Zhang et al., 2025) utilizes multiple graph convolution kernels, while scGNN (Wang et al., 2021) and scSimGCL (Zhang et al., 2024) iteratively refine the cell graph to reduce noise. scTPF (Mrabah et al., 2023) explores the interaction between local and global latent configurations. However, due to the low-quality of scRNA-seq data, similarity metrics often fail to accurately reflect cell groupings, limiting clustering performance. Transformer-based methods, such as scGraphformer (Fan et al., 2024), TOSICA (Chen et al., 2023) and single-cell transformer (Szałata et al., 2024) treat cells as tokens in a sequence and employ a Transformer/Graph Transformer network to capture interactions among cells through the self-attention mechanism. It learns the relationships between any cell pairs, resulting in high computational costs and limited scalability to large-scale scRNA-seq datasets.

### 2.2. Large-scale scRNA-seq Clustering

Mini-batch processing is an effective strategy for handling large-scale scRNA-seq data. Deep embedding–based methods, which encode each cell independently, are naturally suitable for this strategy. For example, scDCC (Tian et al., 2021), scMDC (Lin et al., 2022), IDEC (Guo et al., 2017), and scVAE (Grønbech et al., 2020) encode each cell using fully connected neural networks. Due to the intrinsic noise and sparsity of scRNA-seq data, such independent encoding fails to capture inter-class differences. MetaQ (Li et al., 2025) improves scalability by clustering meta-cells, but overlooks fine-grained cellular heterogeneity. To reduce the complexity of Transformers, several method employ low-rank approximations (Wang et al., 2020; Wu et al., 2022), allowing them to efficiently process long sequences. However, in large-scale scRNA-seq clustering tasks, the sequence length can become excessively large, limiting the effectiveness of these methods. To alleviate this issue, some methods also adopt mini-batch training strategy and construct graphs within each batch (Zhang et al., 2024; Fan et al., 2024). However, this strategy neglects relationships beyond the batch, resulting in the loss of global structural information, which degrades clustering performance. To mitigate this limitation, we design a BGFormer-based clustering model, where shared global tokens enable efficient modeling of long-range interactions across all cells with linear complexity.

## 3. Preliminaries

### 3.1. Notations

Given a scRNA-seq dataset comprising $n$ cells and $d'$ genes, we represent the raw data as a gene expression matrix $\hat{X} \in \mathbb{R}^{n \times d'}$, where $x_{i,j}$ denotes the expression count of gene $j$ in cell $i$. Due to the high dropout rate in gene expression profiles, data filtering and quality control are performed to retain highly variable genes. After log transformation and normalization, the processed data are represented as $X \in \mathbb{R}^{n \times d}$ for downstream tasks, where $d$ denotes the number of selected top-ranked genes.

### 3.2. Transformer-based scRNA-seq Clustering

The goal of scRNA-seq clustering is to ensure that similar cells have the same predictions in an unsupervised manner.

Graph-based clustering methods are formulated as:

$$\hat{Y} = f(AX), \qquad (1)$$

where $A$ denotes the similarity matrix among cells, and $\hat{Y}$ represents the predicted labels. As widely used backbone architecture, Transformers have been employed to implement this process. As shown in Fig. 2(a), the self-attention mechanism in Transformers automatically learns the similarity matrix as:

$$A = softmax(\frac{QK^T}{\sqrt{d_k}}), \qquad (2)$$

where $d_k$ is the dimensionality of $Q$, $Q = XW_Q$ and $K = XW_K$. Here, $W_Q$ and $W_K$ are learnable projection matrices that map cell information into a low-dimensional space, reducing the impact of low-quality data and allowing the model to adaptively learn the similarity matrix during training. $X$ represents the sequence of all cells, with each cell treated as an individual token. The matrix $A$ computed in Eq. (2) can be interpreted as a fully connected graph, where cell pairs with higher similarity are assigned greater weights. Cell information is then propagated over the matrix $A$, which is formulated by:

$$\hat{Z} = AV, V = XW_V, \qquad (3)$$

where $W_V$ is learnable parameters, $\hat{Z}$ represents the learned cell embeddings. To perform clustering, existing methods typically introduce a set of learnable cluster centroids $\{\mu_j\}_{j=1}^K$ and adopt the Deep Embedded Clustering (DEC) objective (Xie et al., 2016). The DEC loss is defined as the Kullback–Leibler (KL) divergence between an auxiliary target distribution and the soft assignment distribution, which is then formulated as:

$$\mathcal{L}_c = KL(P||Q) = \sum_i \sum_j p_{ij} log \frac{p_{ij}}{q_{ij}}, \qquad (4)$$

where $p_{ij}$ is the soft assignment probability of cell $i$ and cluster centroid $j$, and $q_{ij}$ is the corresponding target distribution. After training, the final cluster label for each cell is obtained by selecting the centroid with the highest assignment probability.

While the above method is effective for scRNA-seq clustering, it becomes impractical for large-scale datasets. The self-attention mechanism involves all-to-all interactions, resulting in both computational and memory complexities of $\mathcal{O}(n^2)$. As the number of cells $n$ grows, the costs of training and inference become increasingly prohibitive. This motivates the development of more efficient attention mechanisms to handle such large datasets.

# 4. Methods

We propose a **B**ipartite **G**raph Trans**Former**-based clustering model (BGFormer), which calculates the similarity for

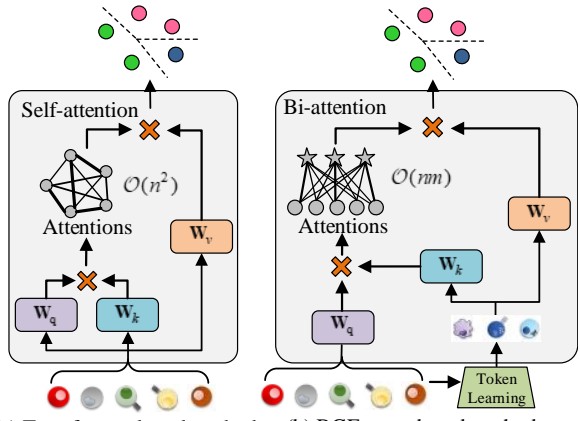

(a) Transformer-based method    (b) BGFormer-based method

*Figure 2.* Comparison of the attention mechanism in traditional Transformer and BGFormer-based clustering model.

clustering using a bipartite graph structure. As shown in Fig. 2(b), BGFormer introduces a set of learnable anchor tokens $U = [u_1, u_2, \ldots, u_m]$, where $u_i$ denotes the $i$-th anchor token and $m$ is the total number of anchors. These anchor tokens serve as reference points to construct the bipartite graph between cells and anchors, enabling computational costs that scale linearly with the number of cells. As a result, BGFormer can be efficiently applied to large-scale scRNA-seq data. However, the design of BGFormer poses two key challenges:

- How can we effectively learn anchor tokens that provide meaningful global information for clustering?

- How can we design a scalable and efficient bipartite graph attention mechanism for clustering?

To address these challenges, the framework of BGFormer is designed as shown Fig. 3, consisting of two key modules, each corresponding to one of the challenges: the anchor token learning and the bipartite graph attention.

## 4.1. Anchor Token Learning

To capture global information, anchor tokens are required to contain the information of all cell. This is achieved by optimizing the tokens through reconstruction of the original cell expression profiles. Specifically, we first encode the input cells into the anchor token space as:

$$H = W_e X + b_e, \qquad (5)$$

where $W_e$ and $b_e$ are the parameters of the encoder, the $i$th row of $H$, $h_i$, is the embedding of cell $i$. The cell embeddings are then mapped to one of the anchor tokens, which is formulated as:

$$j^* = \arg \max_{j \in \{1,\ldots,M\}} \frac{h_i u_j^\top}{\|h_i\|_2 \|u_j\|_2}, \quad u_i^* = u_{j^*}, \qquad (6)$$

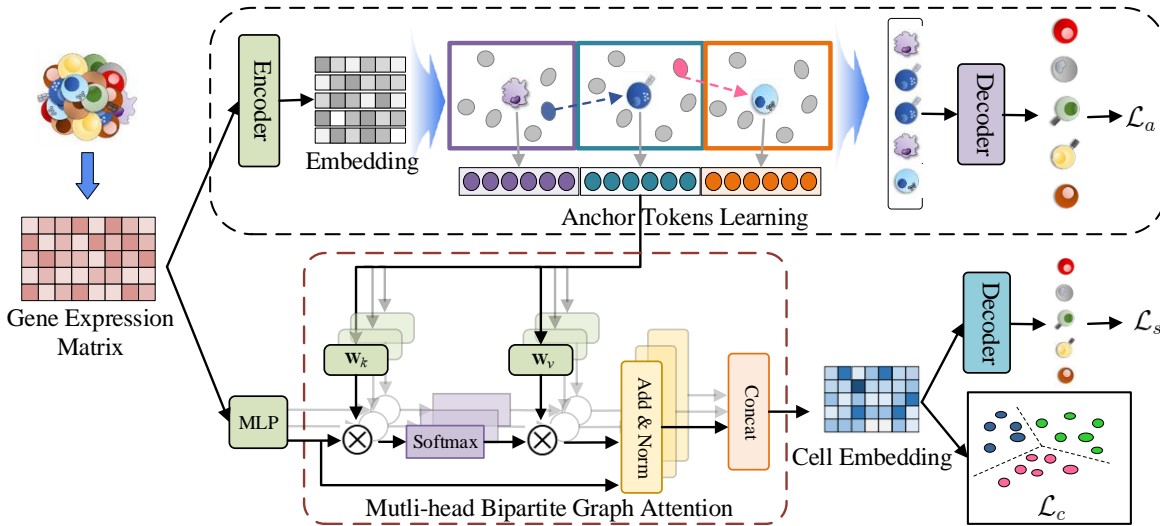

*Figure 3.* The framework of BGFormer-based clustering model.

where $\boldsymbol{u}_i^*$ is the nearest anchor token for cell $i$ in the anchor token space. We feed $\boldsymbol{u}_i^*$ to a decoder to reconstruct the raw cell expression profiles to optimize $\boldsymbol{U}$, which is denoted as:

$$\boldsymbol{h}_i^d = \boldsymbol{W}_d \boldsymbol{u}_i^* + \boldsymbol{b}_d, \quad (7)$$

$\boldsymbol{W}_d$ and $\boldsymbol{b}_d$ are parameters of the decoder. Due to the discrete, over-dispersed, and sparse nature of scRNA-seq data, directly reconstructing the expression matrix is challenging. Following prior work based on the zero-inflated negative binomial (ZINB) distribution (Yu et al., 2022), we estimate output mean $\theta_i$, inverse variance $\mu_i$, and dropout probability $\pi_i$ from the anchor tokens:

$$\begin{aligned} \pi_i &= sigmoid(\boldsymbol{W}_\pi \boldsymbol{u}_i^* + \boldsymbol{b}_\pi), \\ \theta_i &= softplus(\boldsymbol{W}_\theta \boldsymbol{u}_i^* + \boldsymbol{b}_\theta), \\ \mu_i &= exp(\boldsymbol{W}_\mu \boldsymbol{u}_i^* + \boldsymbol{b}_\mu), \end{aligned} \quad (8)$$

where $\boldsymbol{W}_*$ and $\boldsymbol{b}_*$ are parameters matrices. Then, the reconstruction loss function is defined as the negative log-likelihood under the ZINB model:

$$\mathcal{L}_d = -\frac{1}{N} \sum_{i=1}^N log(ZINB(\hat{\boldsymbol{x}}_i | \pi_i, \mu_i, \theta_i)), \quad (9)$$

where $\boldsymbol{x}_i$ is the $i$-th of gene expression matrix $\hat{\boldsymbol{X}}$. To further encourage the anchor tokens to capture the semantics of the cell embeddings, we introduce a commitment loss:

$$\mathcal{L}_{com} = \frac{1}{N} \sum_{i=1}^N ||\boldsymbol{h}_i - \boldsymbol{u}_i^*||_2^2. \quad (10)$$

The total loss function for anchor token learing is:

$$\mathcal{L}_a = \mathcal{L}_d + \mathcal{L}_{com}. \quad (11)$$

## 4.2. Bipartite Graph Attention

We propose a novel bipartite graph attention (Bi-attention) mechanism that constructs a bipartite graph between cells and anchor tokens. The similarity matrix in the bipartite graph is calculated by:

$$\begin{aligned} \boldsymbol{B} &= softmax(\boldsymbol{X}\boldsymbol{W}_p(\boldsymbol{U}\boldsymbol{W}_k)^T), \\ \boldsymbol{Z}_{out} &= \boldsymbol{B}\boldsymbol{U}\boldsymbol{W}_v, \end{aligned} \quad (12)$$

where $\boldsymbol{W}_p$ is used to map cells into low-dimensional embeddings, $\boldsymbol{W}_k$ and $\boldsymbol{W}_v$ are learned parameter matrices that map the anchor tokens $\boldsymbol{U}$ into the key and value spaces, respectively, and $\boldsymbol{Z}_{out}$ is the output representation after aggregating information. Here, $\boldsymbol{B}$ is the attention matrix measuring the similarity between cells and anchor tokens, which serves as the weighted adjacency matrix of the bipartite graph.

To further enhance the capacity of the model to capture diverse interactions, we extend the bipartite attention in Eq. (12) to a multi-head setting, where each head constructs an independent bipartite graph between cells and anchor tokens in a distinct representation subspace:

$$\begin{aligned} \boldsymbol{B}^{(i)} &= softmax(\boldsymbol{X}\boldsymbol{W}_p^{(i)}(\boldsymbol{U}\boldsymbol{W}_k^{(i)})^\top), \\ \hat{\boldsymbol{Z}}_{out}^{(i)} &= \boldsymbol{B}^{(i)}\boldsymbol{U}\boldsymbol{W}_v^{(i)}, \\ \boldsymbol{Z}_{out} &= concat(\hat{\boldsymbol{Z}}_{out}^{(1)}, \hat{\boldsymbol{Z}}_{out}^{(2)}, \ldots, \hat{\boldsymbol{Z}}_{out}^{(l)}), \end{aligned} \quad (13)$$

where $\boldsymbol{W}_p^{(i)}$, $\boldsymbol{W}_k^{(i)}$ and $\boldsymbol{W}_v^{(i)}$ are the projection matrices of the $i$-th head, $\boldsymbol{B}^{(i)}$ denotes the corresponding attention matrix, and $l$ denotes the number of heads. Each head learns a distinct bipartite structure between cells and anchor tokens, enabling the model to capture heterogeneous and complementary relationships between cells and anchor tokens. The

final embeddings can be obtained by

$$\boldsymbol{Z} = \boldsymbol{Z}_{out} + \boldsymbol{X}\boldsymbol{W}_c, \tag{14}$$

where $\boldsymbol{Z}$ represents the learned embeddings for clustering, and $\boldsymbol{W}_c$ denotes the learnable parameters.

### 4.3. scRNA-seq Clustering

To optimize the model parameters, we define the loss function as:

$$\mathcal{L} = \mathcal{L}_s + \mathcal{L}_c + \mathcal{L}_a, \tag{15}$$

where $\mathcal{L}_s$ denotes the self-supervised loss for learning discriminative representations, $\mathcal{L}_c$ is the DEC (Xie et al., 2016) loss that encourages cluster-friendly embeddings (defined in Eq. 4), and $\mathcal{L}_a$ is the reconstruction loss for learning anchor tokens (defined in Eq. 11). Here, self-supervised loss is formulated as a reconstruction objective (similar to Eq. 9), in which each cell is reconstructed from its learned embedding $\boldsymbol{Z}$. The BGFormer algorithm and its workflow are described in detail in Appendix A. The source code of our model is available at the link [1].

When applying BGFormer to large-scale scRNA-seq datasets, we adopt a mini-batch training strategy. Under this setting, conventional self-attention degenerates into a local attention mechanism, as each cell can only attend to other cells within the same batch. In contrast, the proposed bipartite graph attention is well suited to large-scale datasets. The learnable anchor tokens are shared across all mini-batches and are optimized to encode global information from the entire dataset. Consequently, interactions between cells and anchor tokens can effectively approximate aggregation over all cells, even under mini-batch training. This design allows bipartite graph attention to preserve global contextual information, making BGFormer both scalable and effective for large-scale scRNA-seq analysis.

## 5. Theoretical Analysis

When processing large-scale data, an effective solution is to employ a mini-batch training strategy. In this section, we provide a theoretical analysis to demonstrate that our proposed bipartite graph attention can effectively approximate self-attention within each mini-batch.

**Theorem 5.1.** *For any $\boldsymbol{Q}_b \in \mathbb{R}^{n' \times d}$ and $\boldsymbol{K}, \boldsymbol{V} \in \mathbb{R}^{n \times d}$, for any column vector $\boldsymbol{\omega} \in \mathbb{R}^n$ of matrix $\boldsymbol{V}$, there exists a low-rank matrix $\tilde{\boldsymbol{A}}_b \in \mathbb{R}^{n' \times n}$ such that*

$$Pr(\|\boldsymbol{A}_b\boldsymbol{\omega}^T - \tilde{\boldsymbol{A}}_b\boldsymbol{\omega}^T\| < \epsilon\|\boldsymbol{A}_b\boldsymbol{\omega}^T\|) > 1 - o(1), \tag{16}$$

*where $\boldsymbol{A}_b = softmax(\boldsymbol{Q}_b\boldsymbol{K}^T/\sqrt{d_k})$ is the attention matrix, $n'$ is the number of cells in a batch, $\epsilon > 0$ is the error.*

[1]https://github.com/graphlearning1/
BGFormer

We prove this result using the Johnson-Lindenstrauss (JL) Lemma (Johnson & Lindenstrauss, 1984). Let $\tilde{\boldsymbol{A}}_b = \boldsymbol{A}_b\boldsymbol{R}^T\boldsymbol{R}$, where $\boldsymbol{R} \in \mathbb{R}^{m \times n}$ is a random matrix whose entries are drawn independently from $N(0, 1/m)$. According to the JL lemma, for any column vector $\boldsymbol{\omega} \in \mathbb{R}^n$ of matrix $\boldsymbol{V}$, when $m = 5log(n')/(\epsilon^2 - \epsilon^3)$, we have:

$$Pr(\|\boldsymbol{A}_b\boldsymbol{R}^T\boldsymbol{R}\boldsymbol{\omega} - \boldsymbol{A}_b\boldsymbol{\omega}\| \le \epsilon\|\boldsymbol{A}_b\boldsymbol{\omega}\|) > 1 - o(1). \tag{17}$$

Appendix B shows more details. Thus, a low-rank matrix can approximate the attention matrix $\boldsymbol{A}_b$ within an error of $\epsilon$. While SVD-based low-rank approximations offer a feasible solution for approximating the attention matrix $\boldsymbol{A}_b$, the computational cost of performing SVD for each self-attention matrix can be prohibitive. To address this issue, we learn a bipartite graph attention matrix $\boldsymbol{B}$ based on anchor tokens, which approximate the self-attention mechanism.

**Theorem 5.2.** *Let $\hat{\boldsymbol{Z}}$ be the representation produced by self-attention and $\boldsymbol{Z}_{out}$ be the representation produced by the bipartite graph attention, we have:*

$$\|\hat{\boldsymbol{Z}} - \boldsymbol{Z}_{out}\|_F \le \delta, \tag{18}$$

*where $\delta$ is the error bound.*

Following prior work (Xu et al., 2025; Song & Lee, 2023), we simplify the analysis by considering the attention mechanism without the softmax activation. The representation produced by self-attention in Eq. (3) are written as:

$$\begin{aligned}
\hat{\boldsymbol{Z}} &= \boldsymbol{Q}\boldsymbol{K}^T\boldsymbol{V} \\
&\approx \boldsymbol{Q}(\boldsymbol{X}\boldsymbol{W}_K)^T(\boldsymbol{X}\boldsymbol{W}_V) \\
&= \boldsymbol{Q}\boldsymbol{W}_K^T\boldsymbol{X}^T\boldsymbol{X}\boldsymbol{W}_V.
\end{aligned} \tag{19}$$

Similarly, the representation learned by the bipartite graph attention in Eq. (12) can be expressed as

$$\begin{aligned}
\boldsymbol{Z}_{out} &= \boldsymbol{X}\boldsymbol{W}_p(\boldsymbol{U}\boldsymbol{W}_k)^T\boldsymbol{U}\boldsymbol{W}_v \\
&= \boldsymbol{Q}\boldsymbol{W}_k^T\boldsymbol{U}^T\boldsymbol{U}\boldsymbol{W}_v,
\end{aligned} \tag{20}$$

where $\mathbf{U}$ denotes the set of anchor tokens. The reconstruction loss in Eq. (9) guarantees that the anchor tokens $\mathbf{U}$ effectively recover the information from all cells. Formally, we assume there exists a linear decoding matrix $\mathbf{D}$ such that:

$$\|\boldsymbol{X} - \boldsymbol{U}\boldsymbol{D}\|_F \le \delta, \tag{21}$$

where $\delta \ge 0$ denotes the reconstruction error. By substituting $\mathbf{X} \approx \mathbf{U}\mathbf{D}$ into Eq. (19), we obtain:

$$\|\hat{\boldsymbol{Z}} - \boldsymbol{Z}_{out}\|_F \le \delta \tag{22}$$

Therefore, the proposed bipartite graph attention is able to approximate self-attention.

*Table 1.* Clustering performance comparison on scRNA-seq datasets. Bold values denote the best results (%).

| METHOD | CHEN | | BACH | | HRCA | | MRCA | | FETAL-ATLAS | | RATMAP | | ASTROCYTE | |
|---|---|---|---|---|---|---|---|---|---|---|---|---|---|---|
| | ACC | ARI | ACC | ARI | ACC | ARI | ACC | ARI | ACC | ARI | ACC | ARI | ACC | ARI |
| K-MEANS | 16.06 | 37.89 | 56.77 | 48.43 | 41.16 | 26.44 | 53.56 | 46.83 | 46.78 | 5.38 | 53.32 | 39.50 | 39.72 | 13.51 |
| LEIDEN | 51.56 | 37.20 | 90.58 | 87.75 | 49.05 | 8.81 | 83.51 | 78.98 | 52.49 | 34.42 | 61.16 | 45.71 | 57.38 | 30.42 |
| SCICE | 64.00 | 52.81 | 88.25 | 88.35 | OOM | OOM | OOM | OOM | OOM | OOM | OOM | OOM | OOM | OOM |
| IDEC | 18.01 | 23.65 | 54.24 | 77.86 | 51.68 | 19.64 | 57.70 | 58.02 | 40.81 | 31.76 | 46.96 | 32.94 | 44.74 | 20.92 |
| SCMDC | 69.39 | 68.52 | 48.09 | 83.13 | 32.98 | 5.36 | 47.97 | 62.97 | 51.79 | 40.18 | 52.38 | 31.37 | 61.52 | 46.73 |
| SCDCC | 64.84 | 59.30 | 74.47 | 74.20 | 43.74 | 6.43 | 54.61 | 54.11 | 46.20 | 21.03 | 54.09 | 34.56 | 62.15 | 40.52 |
| METAQ | 68.19 | 67.46 | 80.83 | 76.67 | 49.63 | 25.01 | 67.01 | 66.29 | 53.49 | 35.18 | 51.90 | 34.83 | 46.13 | 17.23 |
| SCGNN | 51.24 | 63.98 | 87.43 | 89.11 | OOM | OOM | OOM | OOM | OOM | OOM | OOM | OOM | OOM | OOM |
| SCTAG | 60.52 | 54.82 | 83.74 | 80.15 | OOM | OOM | OOM | OOM | OOM | OOM | OOM | OOM | OOM | OOM |
| CCST | 75.14 | **86.96** | 78.66 | 80.78 | OOM | OOM | OOM | OOM | OOM | OOM | OOM | OOM | OOM | OOM |
| SCTPF | 70.85 | 78.99 | 85.13 | 85.20 | OOM | OOM | OOM | OOM | OOM | OOM | OOM | OOM | OOM | OOM |
| SCGCL | 64.43 | 45.68 | 82.59 | 88.65 | OOM | OOM | OOM | OOM | OOM | OOM | OOM | OOM | OOM | OOM |
| SCG-CLUSTER | 73.26 | 82.40 | 73.26 | 65.94 | OOM | OOM | OOM | OOM | OOM | OOM | OOM | OOM | OOM | OOM |
| SCSIMGCL | 36.32 | 25.24 | 74.93 | 79.53 | 54.24 | 31.05 | 51.38 | 41.51 | 40.90 | 31.06 | 41.16 | 32.97 | 43.40 | 20.50 |
| **OURS** | **80.20** | 86.80 | **91.64** | **90.03** | **68.18** | **48.70** | **89.54** | **90.24** | **60.22** | **43.10** | **64.10** | **52.17** | **70.34** | **50.41** |

## 6. Experiments

### 6.1. Datasets

We evaluate the proposed method on several widely used single-cell RNA sequencing (scRNA-seq) datasets, including Chen (Chen et al., 2017), Bach (Bach et al., 2017), MRCA (Li et al., 2024), HRCA (Li et al., 2023), Fetal-Atlas (Cao et al., 2020), Ratmap (Arduini et al., 2025), and Astrocyte (Schroeder et al., 2025). Most of these datasets contain more than 330,000 cells. Detailed statistical information for each dataset can be found in Appendix C.

### 6.2. Experimental Settings

In our experiments, we set the batch size to 1000. We use the Adam optimizer to optimize the model parameters, with the learning rate selected from {1e-3, 1e-4, 1e-5}. The attention dimension is fixed at 512, with a single attention layer. The number of attention heads is set to 4 for most datasets. The number of anchor tokens is chosen from the set {64, 128, 256, 512, 1024, 2048}. Following (Lin et al., 2022), we select the top 1,500 highly variable genes to construct the feature matrix.

### 6.3. Baselines

To evaluate the performance of BGFormer, we compare it with several representative baseline methods for single-cell clustering. For example, *general clustering baselines* that commonly used in single-cell analysis, including K-means, Leiden (Traag et al., 2019), and scICE (Kim et al., 2025). *Deep embedding methods* encode cells indepen-

dently without explicitly modeling cell–cell interactions, including IDEC (Guo et al., 2017), scDCC (Tian et al., 2021), scMDC (Lin et al., 2022), and MetaQ (Li et al., 2025), where MetaQ is designed for large-scale datasets. *GNN-based methods* construct a $k$-nearest neighbor (kNN) graph over all cells and apply graph neural networks to capture cell–cell relationships, including scGNN (Wang et al., 2021), scTAG (Yu et al., 2022), CCST (Li et al., 2022), scTPF (Mrabah et al., 2023), scGCL (Xiong et al., 2023), scG-cluster (Zhang et al., 2025), and scSimGCL (Zhang et al., 2024). Since scGraphFormer (Fan et al., 2024) requires labeled data during training, it is unsuitable for clustering tasks. Therefore, we exclude it from our comparisons.

We evaluate clustering performance using two standard metrics: accuracy (ACC) (Xie et al., 2016), and adjusted Rand index (ARI) (Hubert & Arabie, 1985). All methods are implemented using the PyTorch framework, and experiments are conducted on an NVIDIA GeForce RTX 4090 GPU with 24 GB of memory.

### 6.4. Clustering Performance

Table 1 summarizes the clustering performance of BG-Former and several baseline models on scRNA-seq datasets. The best values are highlighted in bold, and "OOM" indicates an out-of-memory error. The results indicate that BGFormer almost achieves the highest ACC and ARI across all datasets. Although deep embedding methods can be easily applied to large-scale data, their performance is typically constrained by the poor quality of cell representations learned through independent encoding. GNN-based methods demonstrate superior performance on small-scale

*Table 2.* Training (Train) and inference (Inf) time per epoch (Second).

| | scTPF | | scTAG | | scG-CLUSTER | | scGNN | | scSimGCL | | OURS | |
|---|---|---|---|---|---|---|---|---|---|---|---|---|
| | TRAIN | INF | TRAIN | INF | TRAIN | INF | TRAIN | INF | TRAIN | INF | TRAIN | INF |
| CHEN | 3.32 | 2.33 | 1.16 | 0.21 | 6.66 | 0.98 | 11.18 | 0.65 | 1.31 | 13.13 | 0.81 | 0.31 |
| BACH | 6.66 | 3.25 | 3.11 | 0.25 | 9.98 | 1.14 | 48.25 | 0.95 | 1.91 | 5.47 | 1.62 | 0.47 |
| MRCA | - | - | - | - | - | - | - | - | 23.27 | 166.80 | 9.91 | 6.89 |
| HRCA | - | - | - | - | - | - | - | - | 27.27 | 49.96 | 16.64 | 6.54 |
| FETAL-ATLAS | - | - | - | - | - | - | - | - | 36.86 | 441.41 | 19.74 | 8.08 |
| RATMAP | - | - | - | - | - | - | - | - | 49.37 | 279.01 | 23.07 | 7.73 |
| ASTROCYTE | - | - | - | - | - | - | - | - | 60.25 | 114.64 | 32.53 | 11.83 |

*Table 3.* Training (Train) and inference (Inf) time across different numbers of anchor tokens (Second).

| DATASET | PHASE | 64 | 128 | 256 | 512 | 1024 | 2048 |
|---|---|---|---|---|---|---|---|
| BACH | TRAIN | 1.23 | 1.22 | 1.25 | 1.25 | 1.25 | 1.23 |
| BACH | INF | 0.60 | 0.61 | 0.64 | 0.61 | 0.62 | 0.60 |
| RATMAP | TRAIN | 25.00 | 24.94 | 25.28 | 25.29 | 25.57 | 24.68 |
| RATMAP | INF | 7.49 | 7.89 | 7.47 | 7.45 | 7.95 | 8.05 |

*Table 4.* Complexity analysis of different models.

| | TIME COMPLEXITY | SPACE COMPLEXITY |
|---|---|---|
| scGNN | $\mathcal{O}(n^2 d + nkd^2)$ | $\mathcal{O}(n^2 + knd)$ |
| TRANSFORMER | $\mathcal{O}(n^2 d + nd^2)$ | $\mathcal{O}(n^2 + nd)$ |
| OURS | $\mathcal{O}(nmd)$ | $\mathcal{O}(nm)$ |

datasets. However, they are limited by the high memory cost of kNN graph construction, which can result in out-of-memory errors when applied to large-scale datasets. BG-Former combines the advantages of two kinds of methods. It captures cell relationships by learning a bipartite graph for clustering, where pairwise cell similarities are approximated through the learned relationships between cells and anchor tokens. By avoiding direct interactions among cells, this design preserves strong scalability by enabling independent encoding of individual cells. As a result, BGFormer achieves superior clustering performance and scales effectively to large-scale scRNA-seq datasets. The results of the NMI evaluation metrics are presented in Appendix D.

## 6.5. Efficiency Analysis

To evaluate the efficiency of BGFormer, we compare its runtime per epoch and complexity with several graph-based methods that also capture relationships among cells during clustering. Methods, such as scTAG (Yu et al., 2022), scTPF (Mrabah et al., 2023), scG-cluster (Mrabah et al., 2023), and scGNN (Wang et al., 2021), require the construction of a graph before model training. scSimGCL dynamically constructs local graphs during training.

The comparison of training and inference time per epoch, shown in Table 2, further underscores the efficiency of BG-Former. The "–" in the table indicates that these methods are unable to scale effectively with large-scale datasets due to their high computational and memory complexities. The result shows that our method significantly outperforms existing models on both small- and large-scale datasets. While scSimGCL can handle large-scale datasets, it requires longer training and inference times. Specifically, BGFormer runs $2\times$ faster than scSimGCL during training and 10-20$\times$ faster during inference. Notably, both training and inference times for BGFormer remain stable as the number of cells increases. These findings indicate that BGFormer is an efficient solution for large-scale scRNA-seq clustering. Table 3 shows the training and inference times for the Bach and Ratmap datasets across different anchor token number (64 to 1024). The results demonstrate that both training and inference times remain stable across increasing dimensions. Since the number of anchor tokens ($\leq 1024$) is much smaller than the number of cells ($\geq$ 10K or $\geq$ 300K), the time overhead introduced by varying the anchor token count can be neglected. Appendix E provides an efficiency analysis of the anchor token number.

Table 4 shows the complex analysis of these methods during embedding learning. Taking scGNN (Wang et al., 2021) as a representative GNN-based method, we present its time and space complexities: $\mathcal{O}(n^2 d + nkd^2)$ and $\mathcal{O}(n^2 + knd)$, respectively, which account for the graph construction. Similarly, the standard Transformer model computes a similarity matrix, leading to time and space complexities of $\mathcal{O}(n^2 d + nd^2)$ and $\mathcal{O}(n^2 + nd)$. These high computational and memory complexities further restrict the applicability of these models to large-scale problems. In contrast, BG-Former models interactions between cells and anchor tokens. Its time complexity is $\mathcal{O}(nmd)$, and the memory complexity is $\mathcal{O}(nm)$. Since $m \ll n$, the overall computational complexity of BGFormer scales linearly with the number of cells, i.e., $\mathcal{O}(n)$. Compared to these methods, BGFormer is better suited for large-scale scRNA-seq clustering.

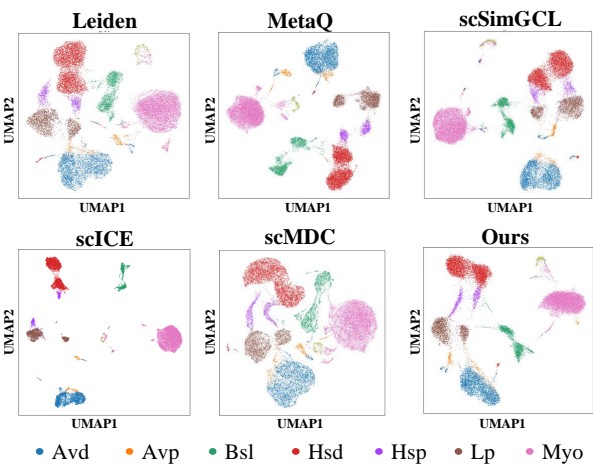

Figure 4. UMAP visualizations of cell embeddings, with colors indicating cell types.

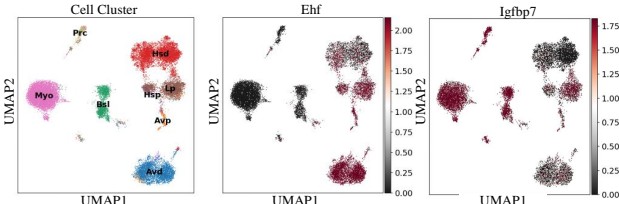

Figure 5. Visualization of expression distribution of the top marker genes.

Table 5. Ablation study on clustering (%).

| DATASET | ACC | | | ARI | | |
|---|---|---|---|---|---|---|
| | W/O $\mathcal{L}_a$ | W/O $\mathcal{L}_s$ | FULL | W/O $\mathcal{L}_a$ | W/O $\mathcal{L}_s$ | FULL |
| CHEN | 64.86 | 60.65 | **80.20** | 55.18 | 50.39 | **86.80** |
| BACH | 83.32 | 82.91 | **91.64** | 80.92 | 83.62 | **90.03** |
| HRCA | 63.94 | 57.00 | **68.18** | 43.39 | 35.17 | **48.70** |
| MRCA | 81.94 | 73.48 | **89.54** | 74.70 | 88.61 | **90.24** |
| FETAL-ATLAS | 52.54 | 46.99 | **60.22** | 32.41 | 28.98 | **43.10** |
| RATMAP | 59.77 | 56.00 | **64.10** | 44.89 | 39.10 | **52.17** |
| ASTROCYTE | 62.44 | 32.59 | **70.34** | 38.72 | 12.72 | **50.41** |

between clusters. In addition, both Alveolar differentiated cells (Avd) and Alveolar progenitor cells (Avp) exhibit high expression of protein-coding genes (e.g., Csn2, Csn1s1 and Wfdc18), supporting the hypothesis that these are secretory alveolar cells. We also analyze the class-wise attention in Appendix F to show the function of anchor tokens.

### 6.7. Ablation Study

To analyze the contribution of each loss component, we compare BGFormer with two variants across all datasets, where each variant omits one part of the total loss during training. Table 5 presents the single-cell clustering results, with 'w/o $\mathcal{L}_a$' indicating the exclusion of Eq. (11) and 'w/o $\mathcal{L}_s$' indicating the removal of self-supervised loss function. The results show that constraining the anchor tokens to encode the key information of all cells enables effective aggregation and propagation of global information, thereby improving clustering performance. Additionally, improving the quality of cell embeddings via self-supervised learning is critical for achieving accurate clustering. Overall, these findings demonstrate that all components of BGFormer are both necessary and effective.

Table 6 shows the ablation study where the learnable anchors are replaced with dynamically updated class centers. As shown in the table, performance consistently dropped across datasets, in terms of both ACC and ARI, demonstrating that independently learned anchors result in better clustering performance. The reason is that class centers are highly unstable during early training and are sensitive to initialization, which can disrupt the model optimization. In contrast, learnable anchors are independently optimized to capture biologically meaningful global representations, ensuring stable training and reliable clustering.

### 6.8. Parameter Analysis

We investigate the effect of varying the number of anchor tokens, denoted as $m \in \{64, 128, 256, 512, 1024, 2048\}$, on model performance. As shown in Fig. 7, the number of an-

### 6.6. Visualization

To evaluate the discrimination of the learned representations, we perform a UMAP visualization (McInnes et al., 2018) on the Bach dataset. In the visualization, cells with the same ground-truth labels are assigned the same color. We compare our model with some representative models from three kinds of methods. As shown in Fig. 4, our model produces well-separated and coherent clusters, demonstrating the effectiveness of the learned representations in distinguishing different cell types. In contrast, scICE demonstrates notable misclassification, while other methods suffer from cellular admixture and cluster overlap. Fig. 5 provides UMAP visualizations colored by marker gene expression on the Bach dataset, which show clear cluster-specific expression patterns of the top marker genes. The results validate high expression of Igfbp7 in basal cells and high expression of Ehf in the secretory lineage (Bach et al., 2017).

Fig. 6 presents a dot plot of the top genes ranked by mean expression across across the predicted clusters produced by BGFormer on the Bach dataset, where dot size and color denote the fraction of expressing cells and the mean expression level, respectively. The varying dot sizes between clusters indicate that BGFormer has successfully distinguished the cell populations, highlighting that it captures the heterogeneity

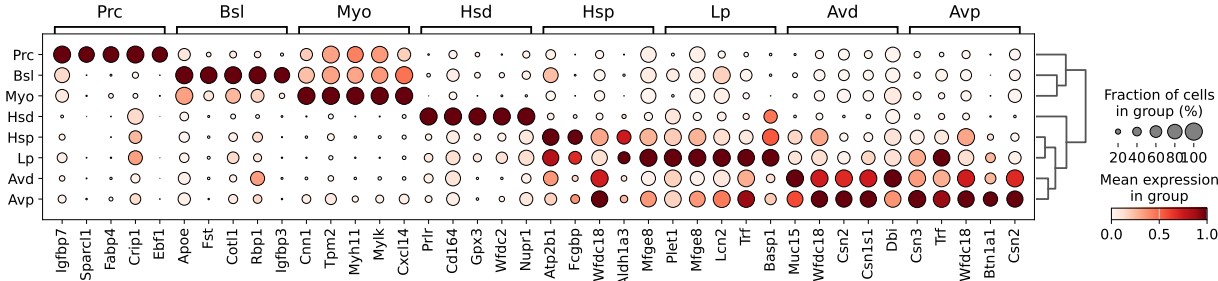

*Figure 6.* Dot plots showing the expression of genes on the Bach dataset. The y-axis represents the identified cell clusters, and the x-axis shows the top-ranked marker genes (top 5 genes per cluster) obtained from differential expression analysis.

*Table 6.* Performance comparison with class centers replacing anchor tokens.

|  |  | CHEN | BACH | HRCA | MRCA | FETAL-ATLAS | RATMAP | ASTROCYTE |
|---|---|---|---|---|---|---|---|---|
| ACC | CLUSTER | 70.30 | 87.84 | 48.28 | 77.25 | 52.23 | 56.01 | 67.01 |
|  | OURS | **80.20** | **91.64** | **68.18** | **89.54** | **60.22** | **63.10** | **70.34** |
| ARI | CLUSTER | 68.54 | 87.63 | 24.50 | 71.65 | 35.93 | 41.89 | 39.01 |
|  | OURS | **80.20** | **90.03** | **48.70** | **90.24** | **43.10** | **52.17** | **50.41** |

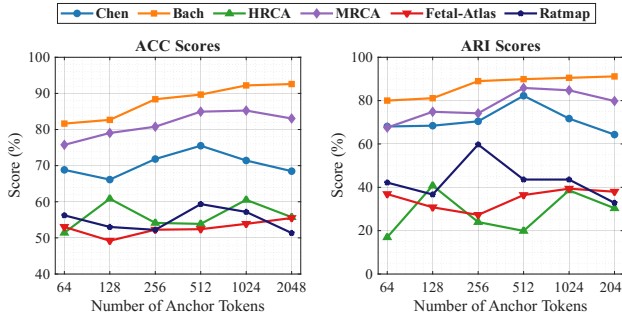

*Figure 7.* Analysis of the number of anchor tokens.

chor tokens significantly influences clustering performance. When the number of anchor tokens is small, the anchors fail to adequately represent the complexity and heterogeneity of large scRNA-seq datasets, resulting in suboptimal clustering performance. As the number of anchor tokens increases, performance steadily improves, indicating that richer anchor representations provide more informative global references for cell embeddings. However, when the number of anchor tokens becomes too large, redundant information is introduced, which can impair model performance. The optimal number of anchors lies within the range of 512 to 1,024. For smaller datasets, 512 anchors are recommended, whereas for large-scale scRNA-seq data, 1,024 anchors are preferable.

## 7. Conclusion

We propose a scalable and effective bipartite graph transformer-based clustering model (BGFormer) for single-cell RNA-seq data. Unlike existing methods that compute

similarities between all pairs of cells, BGFormer focuses on capturing the relationships between cells and a small set of learnable anchor tokens. Because the number of anchor tokens is much smaller than the number of cells, the computational complexity of BGFormer scales linearly, rather than quadratically, with the number of cells. We optimize the anchor tokens to capture global cellular information. A bipartite graph attention mechanism is proposed to learn a similarity matrix that reflects cell groupings and enhances the separation between clusters. Experiments demonstrate that BGFormer achieves superior clustering performance with high efficiency.

## Acknowledgments

This work is supported by the National Natural Science Foundation of China (No.62432006, U21A20473, 62276159) and the Fundamental Research Program of Shanxi Province (No. 202303021223004).

## Impact Statement

This paper presents work whose goal is to advance the field of Machine Learning. There are many potential societal consequences of our work, none which we feel must be specifically highlighted here.

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

## A. Algorithmic Details

Algorithm 1 presents a detailed workflow of BGFormer.

---

**Algorithm 1** BGFormer-based clustering for scRNA-seq data

---

**Input:** Gene expression matrix $\hat{X} \in \mathbb{R}^{n \times d'}$, processed gene matrix $X \in \mathbb{R}^{n \times d}$, number of training epoch $\tau$.
**Output:** Predicted cluster labels $\hat{y}$.
Initialize model parameters, class center $C$ and anchor tokens $U$.
**for** $i = 1$ **to** $\tau$ **do**
    Learning cell embeddings with $U$ using Eq. (13).
    Encode $X$ to obtain representations $H$ using Eq. (5).
    Assign $H$ to anchor tokens using Eq. (6).
    Obtain embedding $Z$ for clustering using Eq. (14).
    Calculate the total loss $\mathcal{L}$ by Eq. (15).
    Update model parameters, cluster centers $C$ and anchor tokens $U$ by minimizing $\mathcal{L}$.
**end for**
Calculate the assignment probabilities of each cell to cluster centroids.
Assign the final label $\hat{y}$ by selecting the centroid with the highest probability.
return $\hat{y}$

---

## B. Proof of Theorem 1

The main proof idea is based on the distributional Johnson–Lindenstrauss lemma (Johnson & Lindenstrauss, 1984).

**Lemma B.1.** *Let $R \in \mathbb{R}^{m \times n}$ be a matrix generated from a normal distribution $N(0, 1/m)$, where $1 \le m \le n$. For any $x, y \in \mathbb{R}^n$, we have*

$$Pr(||Rx|| \le (1+\epsilon)||x||) > 1 - e^{-(\epsilon^2 - \epsilon^3)m/4}, \quad (23)$$

$$Pr(||xR^TRy^T - xy^T|| \le (1+\epsilon)||x||) \le 1 - 2e^{-(\epsilon^2 - \epsilon^3)m/4}. \quad (24)$$

Given an attention matrix $A_b$, with error $\epsilon > 0$, it can be approximated by

$$\tilde{A} = A_b R^T R. \quad (25)$$

Therefore, $rank(\tilde{A}) \le rank(A) \le m$.

Based on Eq. (24), we have

$$\begin{aligned}
&Pr(\|\tilde{A}\omega - A_b\omega\| \le \epsilon \|A_b\omega^T\|) \\
&= Pr(\|A_b R^T R\omega - A_b\omega\| \le \epsilon \|A_b\omega^T\|) \\
&\ge 1 - \sum_{a \in A_b} Pr(\|aR^T R\omega - a\omega\| > \epsilon \|a\omega^T\|) \\
&> 1 - 2n' e^{-(\epsilon^2 - \epsilon^3)m/4}.
\end{aligned} \quad (26)$$

Therefore, when $m = 5log(n')/(\epsilon^2 - \epsilon^3)$, for any column vector $\omega \in \mathbb{R}^n$, we have:

$$Pr(\|\tilde{A}\omega - A\omega\| \le \epsilon \|A\omega^T\|) > 1 - o(1) \quad (27)$$

## C. Dataset

Table 7 shows the detailed statistical information of all datasets.

*Table 7.* Summary of the scRNA-seq datasets.

| DATASET | #CELL | #GENES | #GROUP |
|---|---|---|---|
| CHEN | 12089 | 2500 | 46 |
| BACH | 23184 | 1500 | 8 |
| MRCA | 330930 | 21255 | 27 |
| HRCA | 399605 | 34217 | 5 |
| FETAL-ATLAS | 433695 | 47058 | 58 |
| RATMAP | 504278 | 31110 | 33 |
| ASTROCYTE | 597668 | 26431 | 9 |

## D. NMI-Based Clustering Performance Evaluation

Table 8 shows the clustering performance of our method evaluated using NMI, with the best results in bold and the second-best underlined. From the results, our method achieves either the best or second-best NMI on 5 out of 7 datasets, demonstrating its consistent effectiveness.

## E. Efficiency analysis of anchor token

We have included the training and inference times (in seconds) for the Bach and Ratmap datasets across different anchor token number (64 to 1024), as shown in the table below. The results demonstrate that both training and inference times remain stable across increasing dimensions. Since the number of anchor tokens ($\le 1024$) is much smaller than the number of cells ($\ge 10K$ or $\ge 300K$), the time overhead introduced by varying the anchor token count can be neglected.

## F. Class-Level Attention Visualization

To better understand how the multi-head attention mechanism captures relationships between cells and anchor tokens, we visualize the class-level attention weight distributions across different attention heads. Specifically, attention matrices are grouped according to the ground-truth class labels, and the mean attention score is computed for each class. The resulting class-level attention heatmaps are shown in Figure 8. As illustrated in the figure, different classes exhibit clearly distinct attention patterns toward anchor tokens, indicating that anchor tokens capture class-discriminative global information. Moreover, attention patterns within the same

*Table 8.* NMI scores of different clustering methods on various datasets (%)

| METHOD | CHEN | BACH | HRCA | MRCA | FETAL-ATLAS | RATMAP | ASTROCYTE |
|---|---|---|---|---|---|---|---|
| K-MEANS | 57.04 | 47.66 | 34.87 | 65.05 | 17.24 | 63.46 | 37.57 |
| LEIDEN | 75.21 | 84.76 | 20.95 | 88.38 | **76.14** | **68.10** | 61.47 |
| SCICE | 79.29 | 84.47 | OOM | OOM | OOM | OOM | OOM |
| IDEC | 45.33 | 75.95 | 28.50 | 70.46 | 66.08 | 58.76 | 46.90 |
| SCMDC | 69.50 | **90.76** | 16.14 | 71.01 | 49.04 | 52.03 | 49.35 |
| SCDCC | 67.71 | 76.72 | 15.93 | 64.87 | 43.83 | 58.24 | 51.79 |
| METAQ | 80.19 | 85.39 | 29.96 | 83.98 | 70.83 | 61.78 | 50.09 |
| SCGNN | 74.15 | 90.64 | OOM | OOM | OOM | OOM | OOM |
| SCTAG | 76.44 | 80.71 | OOM | OOM | OOM | OOM | OOM |
| CCST | 77.26 | 77.00 | OOM | OOM | OOM | OOM | OOM |
| SCTPF | 80.62 | 83.43 | OOM | OOM | OOM | OOM | OOM |
| SCGCL | 74.25 | 85.56 | OOM | OOM | OOM | OOM | OOM |
| SCG-CLUSTER | 76.32 | 78.03 | OOM | OOM | OOM | OOM | OOM |
| SCSIMGCL | 57.76 | 79.51 | 35.72 | 67.07 | 50.38 | 59.28 | 53.12 |
| OURS | **81.56** | 86.63 | **42.50** | **89.49** | 66.50 | 66.16 | **55.23** |

*Table 9.* Time across different anchor token number (seconds).

| DATASET | PHASE | 64 | 128 | 256 | 512 | 1024 | 2048 |
|---|---|---|---|---|---|---|---|
| BACH | TRAIN | 1.23 | 1.22 | 1.25 | 1.25 | 1.25 | 1.23 |
| | INFERENCE | 0.60 | 0.61 | 0.64 | 0.61 | 0.62 | 0.60 |
| RATMAP | TRAIN | 25.00 | 24.94 | 25.28 | 25.29 | 25.57 | 24.68 |
| | INFERENCE | 7.49 | 7.89 | 7.47 | 7.45 | 7.95 | 8.05 |

class vary substantially across different heads, suggesting that multi-head attention models class-specific relationships from complementary perspectives.

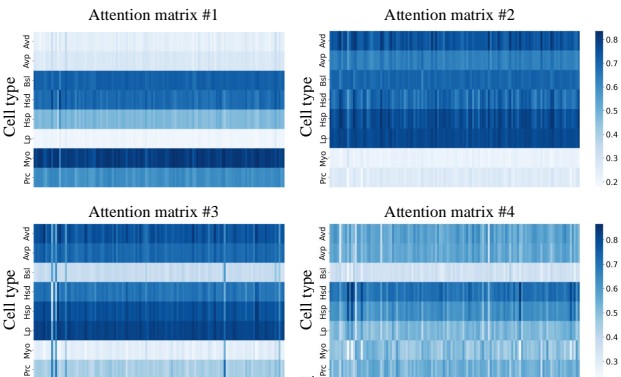

*Figure 8.* Class-wise attention heatmaps of different attention heads.

