# OpenReview forum: "Bipartite Graph Attention-based Clustering for Large-scale scRNA-seq Data"
_ICML.cc/2026/Conference — ICML 2026 regular_

### Official Review · Reviewer_wGHo · 2026-03-05

**Soundness:** 3
**Presentation:** 3
**Significance:** 3
**Originality:** 3
**Overall Recommendation:** 5
**Confidence:** 5

**Summary:**

This paper proposes a Bipartite Graph Transformer-based clustering model (BGFormer) for scRNA-seq data. The model introduces a set of learnable anchor tokens as shared reference points to represent the entire dataset. A bipartite graph attention mechanism is utilized to calculate the similarity between cells and these anchor tokens, reducing the computational complexity to O(n). Finally, experiments on multiple large-scale scRNA-seq datasets are conducted to evaluate the model's clustering performance and computational cost.

**Compliance With Llm Reviewing Policy:**

Affirmed.

**Final Justification:**

The authors have addressed my major concerns, I maintain my positive recommendation.

**Key Questions For Authors:**

1. The anchor token reconstruction relies heavily on the Zero-Inflated Negative Binomial (ZINB) distribution. Have you tested whether this specific noise model generalizes well across different datasets?
2. Does the proposed method inherit the shortcomings of the DEC loss mentioned in the Weaknesses? If so, have the authors implemented any strategies to address this?
3. The stability of the training process remains unclear in the current manuscript. Could the authors provide learning curves demonstrating how the total Loss and ACC fluctuate across epochs during training?

**Limitations:**

Although a sensitivity study is provided, the manuscript lacks a formal discussion on limitations. It is recommended that the authors specify the types of data or scenarios where the model's performance may be constrained.

**Strengths And Weaknesses:**

Strengths:
1. The paper provides a theoretical foundation by proving that the proposed bipartite graph attention can effectively approximate full self-attention within a bounded error.
2. The paper is structurally well-organized and easy to follow.
3. The paper provides detailed algorithm and source code, which enhances the replicability of the proposed model.

Weaknesses:
1. The parameter analysis states that the model is sensitive to the number of anchor tokens. Too few anchors cannot capture the data's heterogeneity, while too many lead to redundancy. However, the authors do not provide a clear criterion or guideline for determining the optimal number of anchors.
2. In the final clustering stage, the model directly employs the DEC loss. However, DEC is known to be sensitive to pre-trained features and initial cluster centroids, making the model prone to falling into local optima.

---

> ### Author Rebuttal · Authors · 2026-03-31
>
> **W1**  Our experimental results (Figure 6) and dataset statistics reveal a trend:
> For small datasets (e.g., Chen, Bach with < 50k cells), 512 anchors achieve peak performance without redundant computation.
> For large datasets (e.g., MRCA, HRCA with > 300k cells), 1024 anchors strike the best balance between clustering accuracy (ACC/ARI) and computational efficiency, as it sufficiently covers complex cellular heterogeneity while avoiding the diminishing returns of 2048 anchors.
> Thus, we recommend 512–1024 anchors as a practical range, with the exact value chosen based on dataset cell count: smaller datasets favor 512, while large-scale scRNA-seq data benefits from 1024.
>
> **W2 and Q2:** Thanks for your comments. Our method mitigates the mentioned limitations of the DEC loss by modeling the relationships between cells and jointly optimizing the representations with a self-supervised loss. Specifically, the self-supervised loss provides an auxiliary signal that enhances the discriminability of representation and reduces over-reliance on initial pseudo-labels, thereby mitigating error propagation and the risk of local optima. Simultaneously, relationship modeling acts as a structural prior to promote more compact intra-cluster aggregation and clearer inter-cluster separation, leading to a more refined and stable feature space. Ablation studies confirm the importance of these two components, as removing either one leads to a significant performance drop, demonstrating their critical role in the clustering process.
>
> **Q1:** The ZINB distribution is widely adopted for modeling scRNA-seq data due to its ability to capture both over-dispersion and excess zeros caused by dropout effects. In BGFormer, the ZINB-based reconstruction loss serves as a probabilistic regularization that encourages anchor tokens and cell representations to preserve biologically meaningful expression patterns.
>
> According to our ablation study, removing the ZINB reconstruction loss consistently leads to performance degradation across all seven datasets, which cover diverse data characteristics. It indicates that this noise modeling mechanism provides stable supervision under diverse data and effectively improves the quality of learned representations.
>
> **Q3:** To clarify the training stability, we provide learning curves for both total loss and ACC, ARI across epochs. These curves demonstrate stable convergence throughout the training process, confirming the robustness of our model under various conditions. (see Figure 4 in https://anonymous.4open.science/r/noi-7B3E/README.md)

---

> > ### Author Rebuttal · Reviewer_wGHo · 2026-04-01
> >
> > Thanks for your rebuttal, the authors have addressed my major concerns. I maintain my positive recommendation.

---

### Official Review · Reviewer_mfVx · 2026-03-07

**Soundness:** 3
**Presentation:** 4
**Significance:** 3
**Originality:** 3
**Overall Recommendation:** 5
**Confidence:** 4

**Summary:**

This paper introduces a novel method called BGFormer for large-scale scRNA-seq clustering. BGFormer is a transformer-based model. To address the quadratic computational complexity of the conventional attention, BGFormer utilizes global learnable anchors to perform similarity learning across samples on a bipartite graph attention mechanism. The authors claim that the BGFormer achieves linear computational complexity with respect to the number of samples, and they demonstrate its effectiveness by comparisons with existing methods on large scale scRNA-seq datasets.

**Compliance With Llm Reviewing Policy:**

Affirmed.

**Final Justification:**

The authors have addressed my concerns, and I keep my positive recommendation.

**Key Questions For Authors:**

1. Could the authors clarify what “train” and “test” refer to in Table 3. It is unclear why a train/test split is needed when the task is purely clustering.


2. Could the authors clarify how the cell labels are predicted in the Algorithm 1?


3. There may be a typo in Line 590. Should it be $rank(\tilde{A}) \leq rank(A_{b}) \leq m$?

**Limitations:**

This work contributes to the biological and medical communities rather than the purely ML community. Discussion of potentially overlooked interpretations or applications in biological and medical contexts is encouraged.

**Strengths And Weaknesses:**

**Strengths:**

The paper is well written and easy to follow. The idea of utilizing global learnable anchors for similarity learning in scRNA-seq clustering through the proposed bipartite graph attention mechanism appears to be novel. The linear computational complexity with respect to the sample number is desirable for increasingly common large-scale scRNA-seq data. Comparisons with various methods on different datasets show the effectiveness of BGFormer in scRNA-seq clustering.

**Weaknesses:**
1. The statement of Theorem 5.2 is informal. It is suggested that the authors either replace it with a remark or revise it to follow a rigorous mathematical formulation.


2. Using only one example for post-hoc analysis in the Visualization section may lead to selective bias. A more systematic analysis with additional cases is preferred.

---

> ### Author Rebuttal · Authors · 2026-03-31
>
> **W1:** Thank you for the helpful suggestion. Following your advice, we revise Theorem 5.2 to provide a more rigorous mathematical formulation by explicitly bounding the approximation error between full self-attention and bipartite graph attention in the Frobenius norm. Specifically, under the assumption that the anchor tokens can linearly reconstruct the original data with bounded reconstruction error, we show that the representations produced by self-attention $\hat{Z}$ and bipartite graph attention $Z_{\text{out}}$ satisfy $|| \hat{Z} - Z_{\text{out}}||_F \leq \delta$. Therefore, the proposed bipartite graph attention is able to approximate full self-attention. We will revise Theorem 5.2 accordingly in the main manuscript.
>
>
> **W2:** Thanks for your comment. We will supplement the visualization section with a more systematic analysis and include additional representative cases in the appendix (see Figure 2 and Figure 3 in https://anonymous.4open.science/r/noi-7B3E/README.md)
>
>
> **Q1:** Thank you for the insightful question. In Table 3, “train” and “test” do not refer to a data split. Instead, they correspond to two different computational stages of the model: training and inference stage. During the training stage, the model parameters are optimized by minimizing the objective function. During the inference stage, the trained model is used to compute the cluster assignment probabilities of cells and produce the final clustering results.
>
> We report both stages because they involve different computational costs. The training time reflects the learning efficiency and resource requirements of the model during the optimization phase, while the inference/testing time determines the model’s practical deployment efficiency and real-time performance. Reporting these two metrics together allows for a comprehensive and thorough evaluation of the model’s overall efficiency.
>
> To avoid confusion, we will revise the terminology by replacing “test” with “inference” and “train” with “Training”, which more accurately reflects the role of this stage.
>
> **Q2:** Thank you for the question. The cell labels in Algorithm 1 are obtained following the clustering strategy described in Section 3.2. Specifically, after training, the model produces the assignment probability of each cell to different cluster centroids in the latent space, and the final cluster label is determined by selecting the centroid with the highest assignment probability. To improve clarity, we will revise the manuscript to explicitly describe the generation of clustering results in the main text.
>
> **Q3:** Thanks for your comment. We will fix the typo in Line 590.

---

> > ### Author Rebuttal · Reviewer_mfVx · 2026-04-02
> >
> > Thank you for the clarification. It has satisfactorily resolved my concerns.

---

### Official Review · Reviewer_yizE · 2026-03-10

**Soundness:** 3
**Presentation:** 3
**Significance:** 3
**Originality:** 3
**Overall Recommendation:** 4
**Confidence:** 4

**Summary:**

This paper introduces a novel bipartite graph Transformer-based clustering model (BGFormer) for large-scale scRNA-seq data analysis. It tackles the key challenge of quadratic computational complexity in Graph Transformer-based methods by employing a bipartite graph attention mechanism with learnable anchor tokens. Experiments on large-scale datasets show that BGFormer outperforms existing GNN and Transformer methods in clustering performance and efficiency.

**Compliance With Llm Reviewing Policy:**

Affirmed.

**Key Questions For Authors:**

(1) Why not use learned class centers as anchors? What are the advantages of learning anchors separately?
(2) The meaning of the axes in Figure 5 is unclear and needs further explanation to ensure proper interpretation of the results.
(3) The experimental results demonstrate that BGFormer achieves excellent clustering performance on both large-scale and small-scale datasets. Could you further discuss the differences in performance improvements across datasets of varying scales and explain the underlying reasons?

**Limitations:**

The experiments were conducted with a fixed number of clusters. Further discussion is needed regarding the case with a variable number of clusters.

**Strengths And Weaknesses:**

Strengths:

(1) The motivation of this paper is clear. The high computational complexity of the self-attention mechanism hinders the scalability of Graph Transformers to large-scale single-cell data analysis tasks. It is meaningful to design more effective attention mechanisms.
(2) The theoretical analysis (Theorems V.1 and V.2) provides justification for why bipartite graph attention can approximate full self-attention. This serves as a strong theoretical foundation for the proposed method.
(3) This paper focuses on modeling global data relationships with low computational complexity, effectively overcoming the limitations of existing methods that primarily capture local information. This advancement significantly enhances the data processing capabilities of Graph Transformers.
(4) The experimental evaluation is comprehensive. Extensive evaluations on large-scale datasets, including clustering performance, visualization analysis, and runtime analysis, demonstrate the effectiveness and scalability of the method.

Weaknesses:
(1) Figure 3 needs further revision. It does not clearly illustrate the role of the outputs from each module, particularly the output of the decoder.
(2) The module name needs to be consistent, as "scBGFormer" is used in some places, while "BGFormer" is used in others.

---

> ### Author Rebuttal · Authors · 2026-03-31
>
> **W1:** Thanks for your comment. We will revise Figure 3 to more clearly illustrate the role of the outputs from each module, including a clearer depiction of the decoder output, to improve the figure’s clarity and interpretability.
>
>
> **W2:** Thanks for pointing this out. We will carefully revise the manuscript to ensure consistent use of the model name BGFormer throughout the text, figures, and tables.
>
>
> **Q1:** Thanks for your question. Using learned class centers directly as anchors would tightly couple the anchor quality with the clustering results. Since class centers are typically obtained based on current feature, their quality can be significantly affected by noise, initialization bias, or inaccurate early representations. This may lead to unstable optimization and error propagation during training.
>
> In contrast, learning anchors separately allows the model to capture global information about the data without relying on the noisy or imperfect early clustering results. This decoupling reduces the risk of errors propagating from early stages to the later representation learning, leading to more stable and reliable clustering.
>
> We conducted an ablation study where the learnable anchors were replaced with dynamically updated class centers. As shown in the table below, performance consistently dropped across datasets, in terms of both ACC and ARI, demonstrating that independently learned anchors result in better clustering performance.
>
> |          |     | CHEN   | BACH   | HRCA   | MRCA   | FETAL-ATLAS | RATMAP | ASTROCYTE |
> |----------|-----|--------|--------|--------|--------|-------------|--------|-----------|
> | Cluster  | ACC | 70.30  | 87.84  | 48.28  | 77.25  | 52.23       | 56.01  | 67.01     |
> | **Ours** | **ACC** | **80.20**  | **91.64**  | **68.18**  | **89.54**  | **60.22**       | **63.10**  | **70.34**     |
> | Cluster  | ARI | 68.54  | 87.63  | 24.50  | 71.65  | 35.93       | 41.89  | 39.01     |
> | **Ours** | **ARI** | **80.20**  | **90.03**  | **48.70**  | **90.24**  | **43.10**       | **52.17**  | **50.41**     |
>
>
> **Q2:** Thanks for your question. Figure 5 visualizes the expression patterns of representative marker genes across predicted clusters. Specifically, the y-axis represents the identified cell clusters, and the x-axis shows the top-ranked marker genes (top 5 genes per cluster) obtained from differential expression analysis. We will revise the figure caption and manuscript to provide a clearer explanation of the axis meanings
>
> **Q3:** Thanks for your question. We observe that BGFormer achieves stable improvements on both small-scale datasets (Chen, Bach) and large-scale datasets (HRCA, MRCA, Fetal-Atlas, Ratmap, Astrocyte), with more pronounced gains on large-scale datasets.
>
> On small-scale datasets, the data structure is easier to learn without significant computational constraints, allowing many methods to achieve competitive performance. However, BGFormer still offers improvements by automatically modeling cell-to-cell relationships and learning more discriminative representations.
>
> On large-scale datasets, gene expression data are typically extremely sparse, making it difficult to learn discriminative representations when cells are modeled independently. Capturing reliable relationships between cells becomes critical for improving clustering performance. However, many existing methods are limited by the high computational complexity of modeling pairwise interactions. BGFormer explicitly models interactions between cells and anchors through bipartite graph attention, which reduces the complexity of modeling pairwise relationships while preserving global information. As a result, BGFormer provides larger performance gains on large-scale datasets.

---

> > ### Author Rebuttal · Reviewer_yizE · 2026-04-06
> >
> > After reading the authors rebuttal, my concerns are addressed. I keep my score unchanged.

---

### Official Review · Reviewer_TQva · 2026-03-12

**Soundness:** 3
**Presentation:** 2
**Significance:** 3
**Originality:** 2
**Overall Recommendation:** 4
**Confidence:** 4

**Summary:**

This paper considers the scRNA-seq clustering problem by introducing a transformer-based method for bipartite graphs. The bipartite graph is constructed with connections between sequence nodes and introduced anchor nodes, helping the aggregation of information while ensuring efficiency. Although effective, current methods generally require O(n^2) time complexity with n being the number of cells, while the proposed method reduces this to O(n), showing significant improvement in efficiency and scalability. The general structure of the paper is well organized, with related literature discussed clearly. Extensive experiments are provided to support the key novelty of the paper.

**Compliance With Llm Reviewing Policy:**

Affirmed.

**Final Justification:**

My concerns have been addressed by the rebuttal. I would like to maintain my positive score.

**Key Questions For Authors:**

Please refer to weaknesses 1-4.

**Limitations:**

Yes.

**Strengths And Weaknesses:**

Strengths:
1.	This paper provides an efficient clustering method for scRNA-seq data, with efficiency and scalability guarantees. The time complexity is O(n), which increases linearly with the number of nodes. The paper is well-organized, with examples and figures to assist with the understanding of the key ideas.
2.	This paper introduces a bipartite graph-based method for clustering by using an attention mechanism to learn the similarity. The anchor nodes have been introduced to allow efficient message communication, which captures global information.
3.	Extensive experiments on large datasets have been conducted to show that the proposed method achieves superior performance and has a lower computational cost compared to existing methods. The UMAP visualization indicates the strong separability between different clusters of scRNA-seqs.
Weakness:
1.	The overall novelty of the proposed framework is generally limited. The major design relies on the introduction of anchor nodes to construct a bipartite graph between cells and anchors, while this solution is already widely used in existing graph learning papers. The anchor nodes targeting graph learning efficiency have been adopted in IDGL (NeurIPS 2020), also, the introduction of similar learnable anchor nodes has been proposed in All-in-one (KDD 2023). The discussion between BGFormer and those methods should be provided to highlight the contribution.
2.	The theoretical analysis mainly shows that the proposed bipartite graph attention can approximate full self-attention. However, this result only suggests that the proposed mechanism can theoretically achieve comparable performance to full attention. It remains unclear why the proposed method achieves the SOTA performance in the experiment. In particular, it would be helpful if the author could analyse which design of the framework is the key to such SOTA performance, and the ablation studies could be extended based on that.
3.	The hyperparameter experiment mainly examines the impact of the anchor token number. It seems that more anchor nodes could lead to better performance. However, the computational cost grows linearly with this number. It would be helpful if the authors could report the corresponding training and inference time to better demonstrate the scalability and efficiency. The current hyperparameter study focuses mainly on the anchor number. It would be clear if more hyperparameters could be evaluated. Moreover, NMI is a commonly used metric in clustering evaluation. Including NMI may make the experiment more comprehensive.
4.	Since this is a framework applied to scRNA-seq data, it would be valuable to provide more biological interpretation of the discovered clusters, such as marker gene analysis.

---

> ### Author Rebuttal · Authors · 2026-03-31
>
> **W1:** Thanks for your comments. Although anchor-based learning has been explored in graph learning, BGFormer differs fundamentally from existing methods in both problem formulation and functional role of anchors.
>
> IDGL (NeurIPS 2020) samples anchors from existing nodes to accelerate graph structure learning, mainly for improving graph construction efficiency. All-in-one (KDD 2023) introduces learnable prompt tokens to align the objectives of pre-trained models with downstream tasks, which are optimized via meta-learning under task supervision.
>
> In contrast, BGFormer treats anchors as latent global tokens that are jointly optimized with cell representations under an unsupervised objective. These anchors are then used in a bipartite graph attention mechanism to automatically learn relationships between cells, addressing the high computational complexity of self-attention mechanism and improving the scalability of Graph Transformers on large-scale data.
>
> **W2:** Thanks for your suggestion. We analyze the key designs behind our SOTA performance.
>
> *a. Avoiding handcrafted similarity graphs.* GNN-based scRNA-seq clustering methods rely on explicit kNN graph construction using predefined similarity metrics, which fail on sparse, noisy scRNA-seq data. BGFormer adaptively learns similarity via bipartite graph attention.
>
> *b. Scalable global dependency modeling.* Graph Transformers require O(n^2) complexity to model pairwise interactions between cells, causing OOM or local attention degradation on large datasets. BGFormer achieves O(n) via learnable anchors shared across mini-batches.
>
> *c. Effectiveness of learnable anchors.* We conduct an ablation study replacing learnable anchors with dynamically updated class centers. As shown in the table below, performance consistently drops across datasets in both ACC and ARI.
>
> |          |     | CHEN   | BACH   | HRCA   | MRCA   | FETAL-ATLAS | RATMAP | ASTROCYTE
> -|-|-|-|-|-|-|-|-|
>  Cluster  | ACC | 70.30  | 87.84  | 48.28  | 77.25  | 52.23       | 56.01  | 67.01
>  **Ours** | **ACC** | **80.20**  | **91.64**  | **68.18**  | **89.54**  | **60.22**       | **63.10**  | **70.34**
>  Cluster  | ARI | 68.54  | 87.63  | 24.50  | 71.65  | 35.93       | 41.89  | 39.01
>  **Ours** | **ARI** | **80.20**  | **90.03**  | **48.70**  | **90.24**  | **43.10**       | **52.17**  | **50.41**
>
> Class centers are highly unstable during early training and are sensitive to initialization, which can disrupt the overall model optimization. In contrast, learnable anchors are independently optimized to capture biologically meaningful global representations, ensuring stable training and reliable clustering.
> We will add these analysis and the ablation study in the Appendix.
>
> **W3:**
> *Scalability analysis of anchor token number.* We have included the training and inference times (in seconds) for the Bach and Ratmap datasets across different anchor token number (64 to 1024), as shown in the table below. The results demonstrate that both training and inference times remain stable across increasing dimensions. Since the number of anchor tokens (<1024) is much smaller than the number of cells (>10K or >300K), the time overhead introduced by varying the anchor token count can be neglected.
>
>  Dataset | Phase     | 64    | 128   | 256   | 512   | 1024  | 2048
> -|-|-|-|-|-|-|-
> Bach    | Train     | 1.23  | 1.22  | 1.25  | 1.25  | 1.25  |  1.23
> Bach    | Inference | 0.60  | 0.61  | 0.64  | 0.61  | 0.62  | 0.60
> Ratmap  | Train     | 25.00 | 24.94 | 25.28 | 25.29 | 25.57 | 24.68
> Ratmap  | Inference | 7.49  | 7.89  | 7.47  | 7.45  | 7.95  | 8.05
>
> *Parameter analysis.* We also analyze the impact of the number of attention heads. The results demonstrate that the model achieves near-optimal performance with 4 attention heads across most datasets (see Figure 1 in https://anonymous.4open.science/r/noi-7B3E/README.md).
>
> *Evaluation Using NMI.* The table shows the clustering performance of our method evaluated using NMI, with the best results in bold and the second-best in italic. From the results, our method consistently achieves the highest or second-highest NMI across most datasets (see Table 1 in https://anonymous.4open.science/r/noi-7B3E/README.md for more results).
>
>  Method   | Chen     | Bach   | HRCA     | MRCA     | Fetal-Atlas | Ratmap   | Astrocyte
> -|-|-|-|-|-|-|-
> Ours     | **81.56** | 86.63 | **42.50** | **89.49** | 66.50       | *66.16*  | **55.23**
>
> **W4:** Thank you for your suggestion. In our paper, we have already presented gene expression analysis in Figure 5. To further assess biological interpretability, we provide UMAP visualizations colored by marker gene expression, which show clear cluster-specific expression patterns of the top marker genes (see Figure 2 in https://anonymous.4open.science/r/noi-7B3E/README.md). The visualization results are provided in the supplementary material.

---

> > ### Author Rebuttal · Reviewer_TQva · 2026-04-02
> >
> > It is a great clarification. My concerns have been resolved. I would like to maintain my positive score.

---

### Decision · Program_Chairs · 2026-04-30

**Decision:**

Accept (regular)

**Comment:**

This paper considers the scRNA-seq clustering problem by introducing a transformer-based method for bipartite graphs. The bipartite graph is constructed with connections between sequence nodes and introduced anchor nodes, helping the aggregation of information while ensuring efficiency. The paper is in a good quality with all positive comments. I will recommend it as Accepted.